# Low coordination number copper catalysts for electrochemical $CO_2$ methanation in a membrane electrode assembly

Yi Xu [1,5], Fengwang Li [2,5], Aoni Xu[2], Jonathan P. Edwards [1], Sung-Fu Hung [2,3], Christine M. Gabardo [1], Colin P. O'Brien [1], Shijie Liu [1], Xue Wang [2], Yuhang Li [2], Joshua Wicks [2], Rui Kai Miao [1], Yuan Liu [2], Jun Li [1,2], Jianan Erick Huang[2], Jehad Abed [2,4], Yuhang Wang [2], Edward H. Sargent [2✉] & David Sinton [1✉]

The electrochemical conversion of $CO_2$ to methane provides a means to store intermittent renewable electricity in the form of a carbon-neutral hydrocarbon fuel that benefits from an established global distribution network. The stability and selectivity of reported approaches reside below technoeconomic-related requirements. Membrane electrode assembly-based reactors offer a known path to stability; however, highly alkaline conditions on the cathode favour C-C coupling and multi-carbon products. In computational studies herein, we find that copper in a low coordination number favours methane even under highly alkaline conditions. Experimentally, we develop a carbon nanoparticle moderator strategy that confines a copper-complex catalyst when employed in a membrane electrode assembly. In-situ XAS measurements confirm that increased carbon nanoparticle loadings can reduce the metallic copper coordination number. At a copper coordination number of 4.2 we demonstrate a $CO_2$-to-methane selectivity of 62%, a methane partial current density of 136 mA cm$^{-2}$, and >110 hours of stable operation.

[1] Department of Mechanical and Industrial Engineering, University of Toronto, Toronto, ON, Canada. [2] Department of Electrical and Computer Engineering, University of Toronto, Toronto, ON, Canada. [3] Department of Applied Chemistry, National Yang Ming Chiao Tung University, Hsinchu, Taiwan. [4] Department of Materials Science and Engineering, University of Toronto, Toronto, ON, Canada. [5] These authors contributed equally: Yi Xu, Fengwang Li. ✉email: ted. sargent@utoronto.ca; sinton@mie.utoronto.ca

The electrochemical $CO_2$ reduction reaction ($CO_2$RR) enables the storage of intermittent renewable electricity while utilising $CO_2$ emissions[1–5]. Methane ($CH_4$) has the largest heating value of 55.5 MJ kg$^{-1}$ (ref. [6]) among $CO_2$RR products, and is the main component of natural gas, well-known for clean and efficient combustion[7,8]. Natural gas provides 24% of global energy, and the infrastructure for $CH_4$ storage, transportation and consumption is established worldwide[9–11]. Thus, the electrochemical conversion of $CO_2$ into $CH_4$ offers a means to close the carbon cycle at a scale relevant to the global carbon challenge (Fig. 1a).

The application of $CO_2$ electrolysis requires catalysts and systems that operate at current densities over 100 mA cm$^{-2}$, exhibit high selectivity, and operate for long lifetimes[12–17]. Prior electrochemical $CO_2$ methanation catalysts have incorporated sputtered copper (Cu) nanoparticles[9], Cu-based alloys[18], covalent triazine framework Cu (ref. [19]) and Cu-complexes[20]. These approaches have increased $CH_4$ selectivities, albeit at low current densities and short run times. $CO_2$ electrolysers incorporating membrane electrode assemblies (MEA) have recently demonstrated significant advancements in reaction stability, current density and scale-up potential[21,22]. The anion exchange membrane also provides a highly alkaline environment at the cathode that has been used extensively to promote C–C coupling and multi-carbon product formation on Cu catalysts[23–25].

Here we report a low coordination Cu catalyst approach for stable and selective electrochemical $CO_2$ methanation in an MEA. We identify, using density functional theory (DFT) calculations, that the reaction energy for the hydrogenation of the *CO intermediate, essential for $CH_4$ generation, is minimised when lowering the global coordination number of Cu from 7.5 to 3.0. To achieve and maintain low coordination number Cu in an MEA, we design a carbon nanoparticle (CNP) moderator strategy. The CNP isolate and prevent the agglomeration of low coordination number Cu clusters formed during the in situ

reduction of a Cu-complex, copper(II) phthalocyanine (CuPc). With a Cu coordination number of 4.2, as verified by in situ extended X-ray absorption fine structure (EXAFS), we achieve a high $CO_2$RR to $CH_4$ Faradaic efficiency (FE) of 62% with a $CH_4$ partial current density of 136 mA cm$^{-2}$ and 110 h of stable electrolysis at 190 mA cm$^{-2}$.

## Results and discussion

**DFT calculations.** We employed a mechanistic $CO_2$RR study to explore the key methanation pathways on Cu catalysts. CO adsorbed on the electrode surface (*CO) is a crucial reaction intermediate toward most $C_1$ and $C_2$ products (Fig. 1b)[26,27]. The adsorbed *CO intermediate faces two diverging pathways leading to different products. In the first option, the *CO intermediate undergoes electrochemical hydrogenation to *CHO, embarking on the $CH_4$ pathway[28]. Alternatively, the *CO can couple in a purely chemical manner (no electron transfer is involved) with another *CO to produce *OCCO, subsequently leading to $C_2$ products, such as $C_2H_4$[29]. This C–C coupling step is enhanced by the highly alkaline conditions of MEA-based electrolyzers[23,24].

We applied DFT calculations to predict the effect of Cu coordination number on hydrogenation and C–C coupling. To accurately represent low values of atomic coordination, adparticles were used to simulate the Cu active sites. Based on these models, we investigated the reaction energies associated with the hydrogenation of the *CO to *CHO (Fig. 1c). The reaction energy for *CO to form *CHO via the hydrogenation step is significantly reduced at low coordination numbers, reaching −0.93 eV at a global coordination number of 3.0, a trend consistent with previous reports[30]. We then investigated the reaction energies associated with C–C coupling of the *CO intermediate toward *OCCO (Fig. 1d). The reaction energies for *CO to form an *OCCO intermediate via C–C coupling did not change significantly with coordination number. Similarly, the adsorption of H$^+$ to *H (on the pathway to hydrogen evolution)

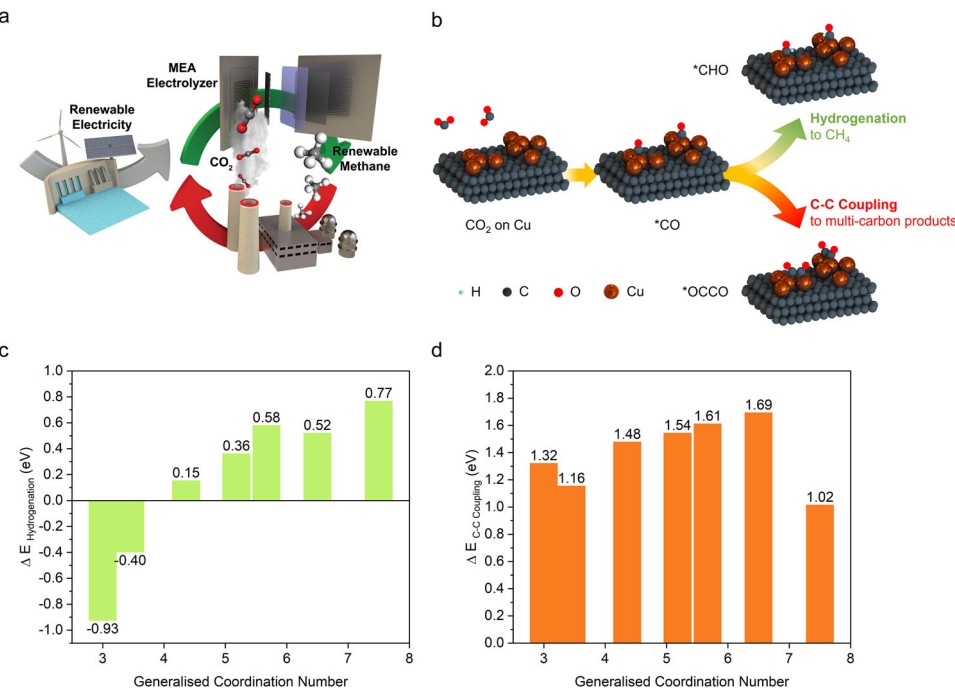

**Fig. 1 $CO_2$RR methanation strategy and DFT calculation. a** Schematic of electrochemical carbon recycling in a membrane electrode assembly (MEA) -based electrolyser for $CO_2$-to-$CH_4$. **b** Schematic of key reaction pathways for $CO_2$RR: hydrogenation to *CHO for $CH_4$ production and C–C coupling to *OCCO leading to $C_2$ generation. **c** Reaction energies for *CO hydrogenation to *CHO on Cu catalysts of various generalised coordination numbers. **d** Reaction energies for *CO coupling to *OCCO on Cu catalysts of various generalised coordination numbers.

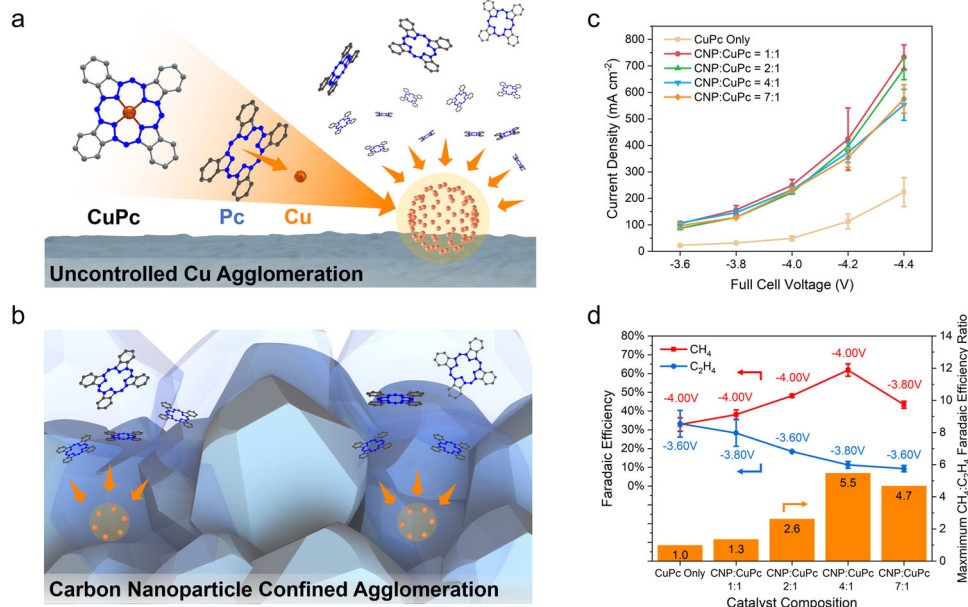

**Fig. 2 Effect of the CNP to CuPc ratio on CO$_2$RR methanation performance.** Schematic of the atomic agglomeration (marked with orange arrows) from CuPc reduction **a** without CNP and **b** with CNP. **c** Current-voltage characteristics for samples with different CNP to CuPc ratios. **d** The maximum FE toward CH$_4$ and C$_2$H$_4$ for samples with different CNP to CuPc ratios when operating between full cell voltages of −3.6 and −4.4 V. Error bars represent the standard deviation of three independent measurements.

is not significantly influenced by the low coordination states when compared to higher coordination states (Supplementary Fig. 1). These findings indicate that low coordination Cu sites promote *CO hydrogenation, presenting an opportunity to boost the selectivity toward CH$_4$ in MEA-based CO$_2$RR systems.

**CO$_2$RR strategy and performance.** To obtain these low Cu–Cu coordination sites, we derived our CO$_2$RR catalyst from CuPc. During the reaction, the Cu atoms in the CuPc molecules reduce from Cu(II) to metallic Cu(0) and then agglomerate into clusters[20,31]. Left uncontrolled, these clusters grew larger than the coordination number predicted by our DFT to be favourable for CH$_4$ production (Fig. 2a). We hypothesised that low coordination number catalysts could be maintained during electrocatalysis with physical confinement within the MEA structure. We designed a CNP moderator strategy to encase and better distribute metallic Cu clusters, thereby resisting Cu agglomeration during the reaction (Fig. 2b).

We formulated catalysts from different mass ratios of CNP:CuPc, including CuPc only, 1:1, 2:1, 4:1 and 7:1. These cathode pre-catalysts were then sprayed on a carbon-based gas diffusion electrode (GDE) typical of high activity CO$_2$RR reactors[32–34]. This spray-based fabrication uses economic precursors and facile preparation, making it compatible with larger electrode fabrication and scaling[35]. Prepared GDEs were coupled with an anion exchange membrane and iridium oxide-based anode for oxygen evolution in the MEA (Methods, Electrode Preparation & Electrochemical reduction of CO$_2$).

Operating the five different samples at full cell voltages between −3.6 and −4.4 V resulted in an exponential increase of current density with cell voltage (Fig. 2c). The sample made from only CuPc exhibited a lower current density at each voltage than the samples with additional CNP. Compared to layers of the low-conductivity CuPc organic framework[36], we posit that the additive CNP improved catalyst utilisation through better electrical contact between neighbouring CuPc molecules. The samples with different CNP ratios exhibited similar current densities at all cell voltages between −3.6 and −4.4 V.

The product distributions obtained during steady-state operation are shown in Supplementary Fig. 2 for all samples. Among the products, CH$_4$ is furthest along the C$_1$ hydrogenation pathway[26]. With C$_2$H$_4$ being the dominant C$_2$ gas product from CO$_2$RR, its selectivity is an indicator of C–C coupling. The selectivities of CH$_4$ and C$_2$H$_4$ provide an indication of the degree of hydrogenation vs. C–C coupling for comparison with our DFT predictions. To account for the effect of the applied voltage on CO$_2$RR selectivity, we plotted the peak FE values for CH$_4$ and C$_2$H$_4$ within the voltage window of −3.6 to −4.4 V (Fig. 2d). We found that increasing the proportion of CNP in the catalyst composition increased CH$_4$ production. The ratio of 7:1 was the exception to this trend, as all CO$_2$RR product FEs, including CH$_4$ and C$_2$H$_4$, decreased while the hydrogen evolution increased (Supplementary Fig. 2b). As the CNP active sites increased relative to that of the Cu, more H$_2$ production is expected because CNP active sites cannot perform CO$_2$RR and instead produce H$_2$ (Supplementary Fig. 2). A lower density of Cu sites thus lowers CO$_2$RR activity. This suggested a trade-off between methanation and reduced CO$_2$RR activity as the ratio of CNP to CuPc was increased. The highest FE toward CH$_4$, 62%, was exhibited by the 4:1 sample at −4.00 V and 220 mA cm$^{-2}$ (liquid product analysis shown in Supplementary Fig. 3). Of the five samples, this 4:1 sample also demonstrated the highest peak CH$_4$ to C$_2$H$_4$ FE ratio of 5.5, highlighting the ability of low coordination Cu states to encourage hydrogenation.

**In situ mechanistic investigations.** To examine the effect of CNP on the Cu coordination number, we investigated the chemical structure at 200 mA cm$^{-2}$, the current density that corresponds to the maxima of CH$_4$ selectivity for the samples studied, with in situ X-ray absorption spectroscopy (XAS). Three representative samples were analysed using this approach: CuPc only, 1:1, and 4:1 ratios of CNP to CuPc, respectively (see Supplementary Note 1).

We carried out in situ Cu K-edge X-ray absorption near edge structure (XANES) spectra to probe the oxidation state of Cu in our CuPc complex. To investigate the structural stability of the

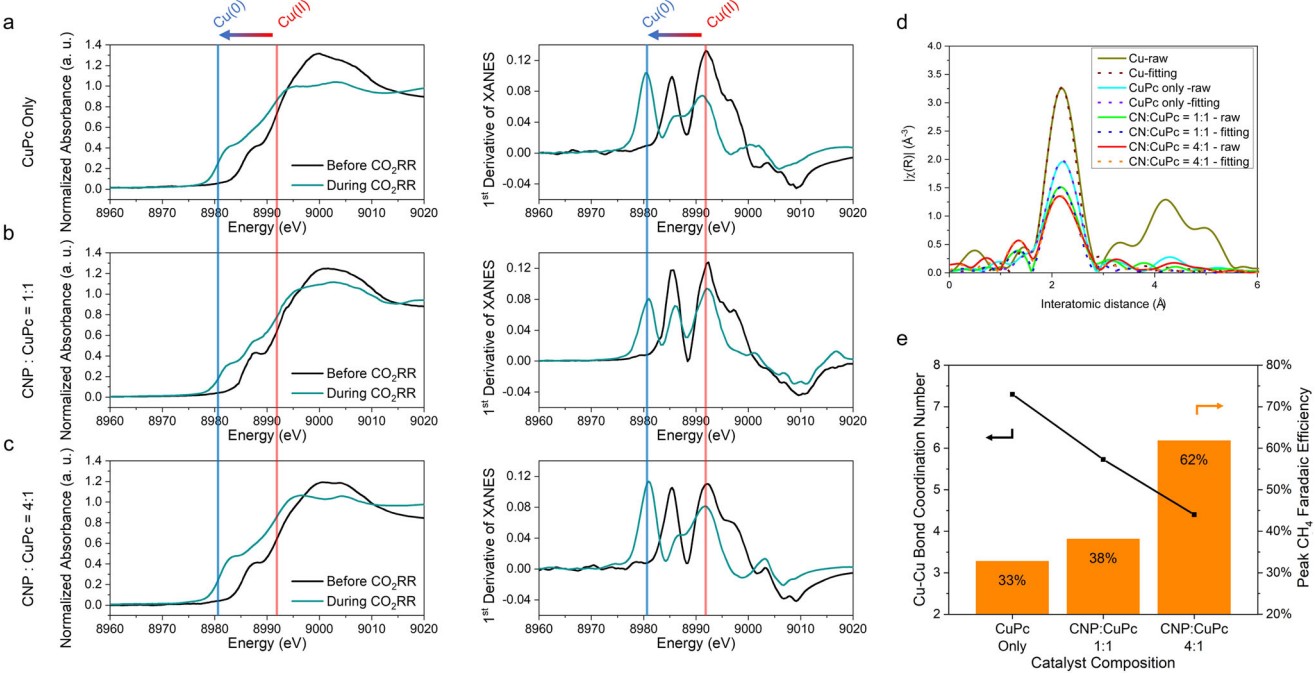

**Fig. 3 In situ sample characterisation under electrocatalytic reaction conditions.** Cu K-edge XANES spectra and first-order derivatives of the XANES spectra (collected at 200 mA cm$^{-2}$ under CO$_2$RR conditions) for the sample containing **a** only CuPc **b** a 1:1 ratio of CNP to CuPc **c** a 4:1 ratio of CNP to CuPc. **d** Fourier-transformed Cu K-edge EXAFS spectra (collected at 200 mA cm$^{-2}$ under CO$_2$RR conditions) and fitting lines for samples containing different ratios of CNP to CuPc. **e** Comparison of the metallic Cu–Cu coordination number determined from EXAFS analysis and methanation selectivity.

lower coordination Cu states, we ran the 1:1 and 4:1 samples for longer times. The first derivatives of the corresponding normalised XANES spectra were also computed to ascertain the spectral absorption peaks (Fig. 3a–c, Supplementary Fig. 5). Before CO$_2$RR, the three groups of samples all presented a characteristic Cu(II) peak at ~8991 eV (ref. [37]), as expected for the CuPc molecule. When a current density of 200 mA cm$^{-2}$ was applied in CO$_2$RR working conditions, the major characteristic peak shifted from the initial Cu(II) peak to the Cu(0) peak located at ~ 8980 eV (ref. [38]). This shift confirmed that most of the Cu within the CuPc catalyst was indeed reduced to metallic Cu(0) during CO$_2$RR, for all sample compositions (Supplementary Table 1).

To investigate the local structure of Cu, we obtained in situ EXAFS spectra under CO$_2$RR conditions (Fig. 3d and Supplementary Fig. 7). The EXAFS spectra of the 4:1 ratio sample stabilised within 50 min of operation with little change between the spectra at 50 and 80 min, demonstrating that the structure was stable once agglomerated (Supplementary Fig. 7d). The fitted metallic Cu–Cu bond scattering path spectra in Fig. 3d were plotted based on fitting parameters shown in Supplementary Table 2.

The metallic Cu–Cu bond coordination number was determined from the fitted scattering paths of the three groups of samples (Fig. 3e). A higher proportion of CNP led to a lower metallic Cu–Cu bond coordination number, confirming the ability of the CNP to limit Cu agglomeration. Combining EXAFS data with CO$_2$RR experimental data indicates that the CO$_2$RR selectivity toward CH$_4$ increased with a decrease of coordination number (Fig. 3e). The highest CH$_4$ FE sample, a 4:1 ratio of CNP vs. CuPc molecules, was found to have a metallic Cu–Cu bond coordination number of ~4.2–much lower than 12, the coordination number of Cu foil (Supplementary Fig. 8). Potentiostatic XAS measurements suggested that CuPc agglomeration was not influenced significantly by the applied potential when the CNP moderator strategy was employed (Supplementary Fig. 9). These results confirm experimentally the DFT prediction that Cu

catalysts with low atomic coordination numbers can boost CO$_2$RR methanation.

**Ex situ mechanistic investigations.** To further investigate the catalytic mechanism, we examined the chemical structure ex situ using X-ray photoelectron spectroscopy (XPS). The highest CH$_4$ selectivity performance sample (4:1 ratio of CNP to CuPc) was analysed and compared to its pre-electrolysis state. Since XPS is an ex situ measurement, we expected some oxidation of the Cu sample during the sample disassembly, preparation, and transport. The deconvolved N 1s peaks demonstrate the N–Cu bond in the CuPc molecular structure was decomposed irrevocably[39] (Fig. 4a and Supplementary Fig. 10 and Supplementary Table 3). The deconvolved Cu 2p peaks show that the CuPc molecules were reduced to the metallic Cu (0) state[39,40], and most do not revert back to CuPc after the electrolysis (Supplementary Fig. 11). The post-electrolysis X-ray diffraction (XRD) results of the samples demonstrated a decrease in the CuPc characteristic diffraction patterns, compared to the pre-electrolysis states, further confirming CuPc decomposition (Supplementary Fig. 12). These XPS and XRD findings support the conclusion of the in situ XAS measurements, namely that metallic Cu is derived during CO$_2$RR and does not revert to CuPc.

Scanning electron microscopy with secondary electrons (SEM) and backscattered electrons (BS), and transmission electron microscopy (TEM) were employed to investigate the morphological changes of the pre- and post-electrolysis catalyst samples. The visual absence of large particle formations in the SEM and BS images of the pre- and post-electrolysis samples suggest no large metallic Cu agglomerates were formed during electrolysis (Supplementary Fig. 13). Energy dispersive X-ray (EDX) mapping and spectroscopy results proved the Cu element is evenly distributed on the GDE (Supplementary Figs. 14, 16 and Supplementary Table 4). STEM/EDX and TEM images taken pre- and post-electrolysis also support the claim that Cu

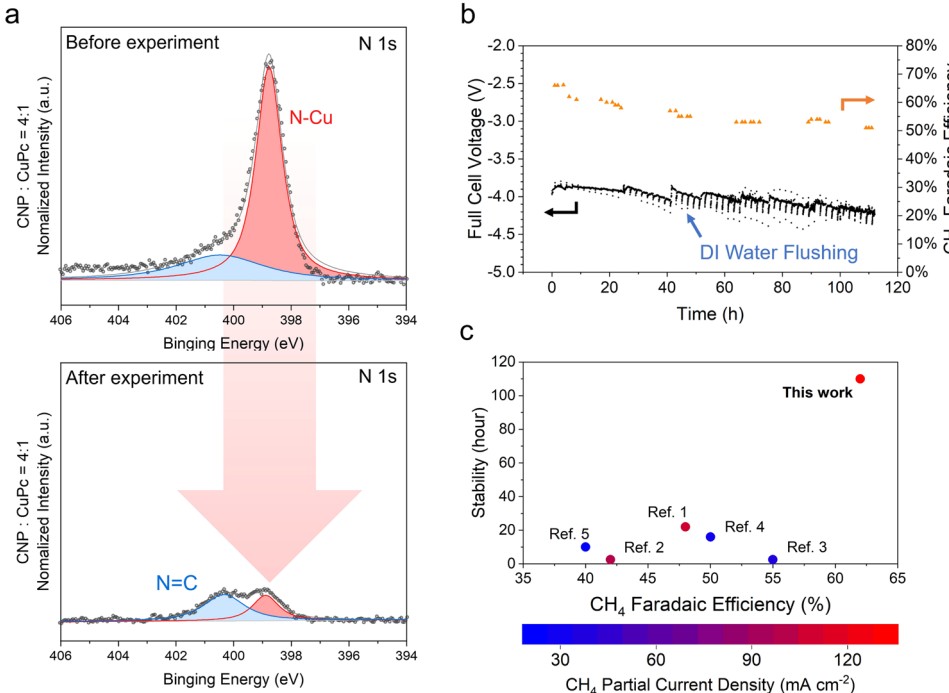

**Fig. 4 Ex situ sample characterisation, stability, and performance comparison. a** Pre-electrolysis and post-electrolysis N 1*s* XPS spectra for a sample containing a 4:1 ratio of CNP to CuPc. **b** Prolonged $CO_2$ electroreduction to $CH_4$ for a sample containing a 4:1 ratio of CNP to CuPc at a fixed current density of 190 mA $cm^{-2}$. **c** Comparison with the previous $CO_2$RR to $CH_4$ product data reports at high current density. The references are from ref. [9]; ref. [52]; ref. [53]; ref. [54] and ref. [55].

nanoclusters (~2–5 nm) were formed during the reaction (Supplementary Figs. 16, 17).

**Stability and performance comparison**. To investigate the electrochemical stability of our low coordination Cu catalyst, we performed extended electrolysis galvanostatically at a current density of 190 mA $cm^{-2}$ (Fig. 4b). To prevent salt accumulation in the gas diffusion layer (GDL) micropores and maintain $CO_2$ mass transport through the GDL, every 2 h we briefly injected DI water into the cathode flow channel (see current fluctuations on Fig. 4b). Every ~24 h additional anode electrolyte, 0.05 M $KHCO_3$, was provided to maintain the original electrolyte volume. The low concentration of electrolyte was chosen to minimise potassium cation crossover and subsequent salt formation[41,42]. Over the course of the experiment, an average $CH_4$ selectivity of 56% was achieved, and the non-*i*R compensated full cell voltage exhibited only a minor change, from −3.9 to −4.2 V. In contrast, applying this strategy in a flow cell configuration with liquid catholyte provided only 2 h of continuous operation prior to flooding (Supplementary Fig. 18), a failure mode typical of these systems[22,43].

We showcase the performance achieved in this work in the context of three metrics essential for industrial implementation of $CO_2$RR: current density, product selectivity, and stability (Fig. 4c). Compared to the highest current density $CH_4$-focused literature, this work outperformed other catalysts in $CH_4$ selectivity and stability. None of the literature reports in Fig. 4c are from MEA-based electrolysers because this work marks the first MEA system capable of selective (e.g., >50%) $CH_4$ production from $CO_2$ (refs. [21,44,45]). The highest FE towards $CH_4$ reported to-date in an MEA is 32%, less than half of the peak methane FE reported in this work[45].

In summary, we studied the critical mechanism pathways for $CO_2$RR methanation and multi-carbon production on Cu catalysts. We found low coordination numbers to be beneficial for reducing the hydrogenation energy requirement toward $CH_4$, counteracting the traditionally low energy requirements for C–C coupling found in MEA systems due to the high basic alkaline conditions found on the cathode. Guided by this finding, we designed a low coordination Cu catalyst, synthesised with CNP additives, which encase and better distribute Cu clusters to prevent excessive agglomeration, for use in an MEA system. This catalyst-system combination converted $CO_2$ to $CH_4$ at a FE of 62% and a partial current density of 136 mA $cm^{-2}$. The same catalyst operated for 110 h at a current density of 190 mA $cm^{-2}$ with an average FE of 56%. The stability of this strategy is a significant advance toward electrochemically derived renewable methane.

## Methods

**DFT calculations**. We performed DFT calculations with the Vienna Ab Initio Simulation Package (VASP) code[46,47]. The exchange correlation energy was modelled by using Perdew-Burke-Ernzerhof (PBE) functional within the generalised gradient approximation (GGA)[48]. The projector augmented wave (PAW) pseudo-potentials[49] were used to describe ionic cores. The cutoff energy of 500 eV was adopted after a series of tests. A Methfessel-Paxton smearing of 0.05 eV to the orbital occupation is applied during the geometry optimisation and for the total energy computations. In all calculations, the atoms at all positions have Hellmann–Feynman forces lower than 0.02 eV $Å^{-1}$ and the electronic iterations convergence was $10^{-5}$ eV using the Normal algorithm. A 4-layer (4 × 4) Cu (111) supercell was built to simulate the exposed surface of Cu while ensuring a vacuum gap of 15 Å. The Cu active sites with various coordination numbers were constructed via creating Cu vacancies on the surface or subsurface. We first ran the stability test for these defective surface models to choose the most stable Cu configurations (Supplementary Table 6 and Supplementary Note 2). Structural optimisations were performed on all slab models with a grid of (3 × 3 × 1) k-point. During the adsorption calculations, the top three layers are fully relaxed while the other layers are fixed at the tested lattice positions. All reaction energy calculations were described in Supplementary Note 3.

**Electrode preparation**. The cathode GDE were prepared by airbrushing catalyst inks with a carrier gas of nitrogen. The catalyst ink was prepared with 30 mL ethanol (Greenfield Global Inc., >99.8%), 150 µL Nafion (Fuel Cell Store D521 Alcohol based 1100 EW, 5 wt%) and catalytic material. The catalytic material

quantities varied by sample. For example, the sample containing only CuPc was made with 14 mg of CuPc (Sigma–Aldrich 546682, >99%). Similarly, The catalytic materials for CNP:CuPc ratios of 1:1, 2:1, 4:1, 7:1 and CNP only were made with 14 mg CNP (Alfa Aesar 39724, 75 m$^2$ g$^{-1}$) + 14 mg of CuPc, 14 mg CNP + 7 mg CuPc, 14 mg CNP + 3.5 mg CuPc, 14 mg CNP + 2 mg CuPc and 14 mg CNP + no CuPc, respectively. The catalyst ink mixtures were sonicated for 5 h, and then sprayed on a gas diffusion carbon paper (Fuel Cell Store Sigracet 39 BC, with microporous layer) with a spray density of 0.22 mL cm$^{-2}$. After airbrushing, the GDE was dried for 24 h at room temperature (~20 °C). The anode electrode was prepared by dip-coating iridium chloride (Alfa Aesar, IrCl3·xH2O 99.8%) on titanium support (0.002" thickness, Fuel Cell Store). Then, IrO$_x$ was formed on the coated electrode by thermal decomposition in air[50].

**Characterisation.** Catalyst surface morphology was imaged by a Hitachi S-5200 SEM at 10 kV and JEOL JEM-2010 HRTEM at 200 kV. Elemental analysis was investigated with field-emission scanning electron microscopy (FE-SEM, JEOL JSM-6700F) equipped energy dispersive X-ray spectroscopy (EDX, Oxford Instrument XMax 150 mm$^2$). The lattice fringes and elemental mapping were collected with field-emission transmission electron microscopy (FE-TEM, JEOL-2100F) equipped energy dispersive X-ray spectroscopy (EDX, Oxford Instrument XMaxN TSR). In situ hard XAS measurements were conducted using a modified flow cell at beamline 9BM of the Advanced Photon Source (APS, Argonne National Laboratory, Lemont, Illinois, United States)[51] and the silicon drift detector at the 44 A beamline of National Synchrotron Radiation Research Center (NSRRC, Hsinchu, Taiwan). XRD measurements were performed on a Rigaku MiniFlex600 G6. XPS measurements were conducted using a Thermo Scientific K-Alpha spectrophotometer with a monochromated Al Kα X-ray radiation source.

**Electrochemical reduction of CO$_2$.** All CO$_2$RR experiments were performed using an MEA electrolyser with an active area of 5 cm$^2$ (Supplementary Fig. 19). During a CO$_2$RR experiment, the aqueous 0.05 M KHCO$_3$ anolyte was circulated through the anode flow channel at a flow rate of 25 mL min$^{-1}$ using a peristaltic pump. An anion exchange membrane (Sustainion X37–50, Dioxide Materials) was used as the solid cathode electrolyte. The CO$_2$ gas flow rate, supplied at a rate of 80 standard cubic centimetres per minute (sccm), was bubbled through water for humidification prior to entering the electrolyser. All voltages reported are full cell voltages without iR compensation.

The CO$_2$RR gas products were analysed in 1 mL volumes using a gas chromatograph (PerkinElmer Clarus 590) possessing a thermal conductivity detector (TCD) and a flame ionisation detector (FID). The liquid products were quantified using nuclear magnetic resonance spectroscopy (NMR). $^1$H NMR spectra of freshly acquired samples were collected on an Agilent DD2 500 spectrometer with dimethyl sulfoxide (DMSO) as an internal standard. For the screening of samples with different CuPc:CNP ratios, gas and liquid samples were taken after two hours of CO$_2$RR to ensure that the system was at steady state. Faradaic efficiency (FE) of CO$_2$RR gas product was calculated by the following equation:

$$FE_{gas} = x_i \times v \times \frac{z_i F P_0}{RT} \times \frac{1}{j_{total}} \times 100\% \tag{1}$$

where $x_i$ is the volume fraction of gas product $i$, $v$ is the outlet gas flow rate in sccm, $z_i$ is the number of electrons required to produce one molecule of product $i$, $F$ is the Faraday Constant, $P_0$ is atmosphere pressure, $R$ is the ideal gas constant, $T$ is the temperature, and $j_{total}$ is the total current.

The FE of CO$_2$RR liquid product was calculated by the following equation:

$$FE_{liquids} = n_i \times \frac{z_i F}{Q} \times 100\% \tag{2}$$

where $n_i$ is the number of moles of liquid product $i$, and $Q$ is the cumulative charge as the liquid products were collected.

## Data availability

The data that support the findings of this study are available from the corresponding author on reasonable request.

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

## Acknowledgements

We acknowledge support from the Natural Sciences and Engineering Research Council (NSERC) of Canada. Support from Canada Research Chairs Program is gratefully acknowledged, as is support from an NSERC E.W.R. Steacie Fellowship to D.S. Y.X. acknowledges NSERC for their support through graduate scholarships. J.P.E. thanks NSERC, Hatch, and the Government of Ontario for their support through graduate scholarships. We acknowledge Centre for Nanostructure Imaging at the University of Toronto and Dr. Ilya Gourevich for sample SEM and TEM characterisation. Thanks to Ms. C.-Y. Chien of the Ministry of Science and Technology (National Taiwan University) for the assistance in FE-TEM and EDS experiments. This research used resources of the Advanced Photon Source, an Office of Science User Facility operated for the U.S. Department of Energy (DOE) Office of Science by Argonne National Laboratory and was supported by the U.S. DOE under Contract No. DE-AC02-06CH11357, and the Canadian Light Source and its funding partners. We thank Dr. Tianpin Wu and Dr. George Sterbinsky from 9BM beamline for assistance in collecting the XAS data at the advanced photon source (APS).

## Author contributions

D.S. and E.H.S. supervised the project. Y.X. designed and carried out all the experiments. F.L. designed and analysed the sample characterisation. A.X. carried out the DFT simulation. Y.X., F.L. analysed the experimental data and prepared the manuscript. C.M.G., C.P.O., S.L. and R.K.M. designed and performed the stability test. S. F.H., F.L., X.W. and J.A. performed and analysed the in situ and ex situ XAS measurements. J.W. performed the XPS measurements. Y.Liu. performed XRD measurements and data analysis. Y.Li., J.L., J.E.H. and Y.W. contributed to data analysis and manuscript polishing. All authors discussed the results and assisted during manuscript preparation.

## Competing interests

The authors declare no competing interests.
