## [Peer Review File · Nature Communications]

Reviewer #1 (Remarks to the Author):

In this article, the authors investigate electrocatalysts for the conversion of CO₂ to CH₄. Specifically, the authors find via computational studies that Cu in a low coordination environment favors the conversion of CO₂ to CH₄ in alkaline media as opposed to multi-C products. The authors then develop a strategy to use carbon nanoparticles to confine Cu to an ultra-low coordination environment in their prepared MEAs. By increasing the amount of C nanoparticles, the authors show that the Cu coordination number can be reduced to 4.2, which hinders the formation of C-C products and promotes CH₄ formation. Such a study of the restructuring of Cu-based site isolated catalysts and its tuning to modulate product distribution is of prime importance for the field, and fully justifies its publication in this journal.

However, this article would require some corrections and additions before the work could be published, mainly regarding the processing and analysis of XAS data. A list of comments and questions can be found below:

1. Page 6, the authors state “The ratio of 7:1 is the exception to this trend, as all CO₂RR product FEs, including CH₄ and C₂H₄, decreased while the hydrogen evolution increased (Supplementary Fig. 2b). We attribute this selectivity shift to a shortage of Cu reaction sites, suggesting a trade-off between methanation and reduced CO₂RR activity as the ratio of CNP to CuPc is increased”

This formulation is not very clear. Are the authors claiming a shift in selectivity originating from a mass-transport limitation of CO₂ to the active sites? This statement may need to be reformulated.

2. The XANES and EXAFS for 4:1 CNP:Cu do not seem to agree so well. The XANES indicate that all of the Cu(II) is reduced to Cu(0) within the first 25 minutes and that edge position remains stable afterwards. This is further highlighted by the first derivative (Figure 3c). The EXAFS, on the other hand, show that the Cu(II) is gradually reduced and a significant amount remains after 25 minutes and there is even still some present after 50 minutes (Supp. Figure 3c). In comparison, the XANES for 1:1 CNP:CuPc also seem to be in disagreement. The XANES (Figure 3b) seem to show a gradual shift from Cu(II) to Cu(0) with time while the EXAFS show that most of the Cu(II) has been converted to Cu(0) within the first 20 min and changes only slightly afterwards (SI Figure 3b).

Could the authors explain these discrepancies? Is it possible that the spectra in Figures 3b and 3c were accidentally switched?

3. Figure 3d shows the fitting after CO₂RR (no potential applied)? Or under which applied potential? This should be clarified in the Figure 3 caption.

4. Page 8, lines 1-2, the authors state “The fitted metallic Cu-Cu bond scattering 1 path spectra in Fig. 3d were plotted based on fitting parameters shown in Supplementary Table 1.”

5. However, this table does not really provide a summary of the fit. What do the CuPc and Cu ratios refer to? This is not explained or described in the text at all. A table of the fitting parameters including the CN, bond length, Debye Waller factor, e_0 , r-factor, etc. used or calculated for each scattering path that was included in the fit should be provided for each sample at each potential. In addition, it appears from that table that the EXAFS fits were carried out assuming only Cu-Cu coordination. Wouldn't the author expect some Cu-C bonds once CNPs are added?

If very small Cu particles are considered, then some evidence of Cu-C coordination from the CNPs would be likely, at least from where they are anchored to the CNP surface; could the author comment on that point?

6. Similarly, I found the experimental methods section to be significantly lacking detail regarding XAS data. A better description of the XAS experiments is absolutely needed since these are such an integral part of this study. The fitting method for the EXAFS data needs to be described: Which software was used, what fitting parameters were used, which parameters were varied or held constant, etc. All EXAFS fitting results need to be included and presented in a table (coordination number, bond length, amplitude reduction factor, debye waller factor, E0, r-factor, etc.). Over what range (k and R) were the data fit? Which scattering paths were used to fit the data? At the moment there is nothing to really indicate the quality of the EXAFS fitting. Ideally, an example of one of the fits in k-space and r-space (with the real or imaginary component) should be shown.

7. The second main point that currently appears unclear from the article is the timeframe of the transformation of the CuPC to Cu nanoparticles. Could the author develop on that point? In particular, how long do CNP:CuPC samples need to be reduced before they reach optimal selectivity for CH₄? Is there a break in period or some preconditioning applied where the CO₂RR is increasing as the catalyst is formed? Or are the data in Figure 2c/d collected immediately after assembling the cell?

8. In the same line, is there a reason that the XAS results are shown for OCC (i.e. 0 min), 3, 6, and 10 min for the in-situ characterization of CuPC while for the preparation of 4:1 CNP:CuPC the first data point is after 25 minutes? Is there anything interesting that happens within the first few minutes that shows the formation of the catalyst? From the XANES it seems that all the Cu(II) is reduced on a shorter time scale for the 4:1 sample.

9. Page 8, line 10-12 the authors state "The XAS characterisation employed here yields an average coordination number within the first ~100 nm of the bulk, and thus provides a conservative (e.g. high) estimate of the coordination number at the surface.

I am not sure what is meant by this statement. How big do the authors expect the Cu particles to be? From the post-electrolysis TEM image (supp. Figure 9) it appears they should be around 2-5 nm, so XAS should be very representative of what is happening at the surface. Anything much larger than 20 nm in diameter would only yield information related to the bulk.

10. The reported electrolyte concentration seems rather low (0.05 M KHCO₃). Could the authors comment on this choice?

11. The XPS figures in the SI (Supp. Figure 5) are very difficult to read because of the color choice. The light grey color used to plot the measured signal is nearly impossible to read against the white background. This needs to be updated to a darker color that is easily visible. The total fit envelope of the data also needs to be shown and compared to the measured signal, not just the fits of the deconvoluted peak components.

12. From these XPS data, I would argue that a significant amount of CuPC remains even after electrolysis, considering that the CuPC peaks in the Cu 2p and N-Cu peaks in the N 1s are clearly visible in all samples after electrolysis, and also in the XRD. A quantitative analysis should be applied to the XPS data if possible in order to identify how much CuPC remains after electrolysis. The incomplete conversion of CuPC to CNP:CuPC also seems to be visible in the EXAFS shown in Supp. Figure 3b and 3c (i.e. the peak remaining around 1.5 Å), which may also contribute to the lower

overall CNs that are derived. Do the authors have a hypothesis on this point?

13. The authors state that “the higher magnification TEM images taken pre- and post-electrolysis also support the claim that no sizable metallic Cu nanoparticles were formed during the reaction while also suggesting that the metallic Cu was in an amorphous state during electrolysis.”

What do the author consider sizeable metallic Cu nanoparticles? Based on the comparison of the TEM images, it seems that some particles in the range of 2-5 nm could be suspected. I recommend adding STEM and EDX mapping to the paper to help with identifying the presence or absence of small Cu nanoparticles. Being able to map the distribution of Cu before and after electrolysis would be highly desirable.

Some additional minor issues:

14. In the SI Figure 2 caption, “(b) The Faradaic Efficiency (FE) toward hydrogen with different CNP to CuPc ratios.” is written twice.

15. Supp. Figure 11 is not referenced in the article despite its interesting content and should be mentioned in the main text.

16. Are the XANES and EXAFS shown in Figure 3 performed in a saturated electrolyte (i.e. under constant CO₂ bubbling)? This should be indicated in the figure caption

17. A better description of the flow cell setup should be added in the experimental part or supplementary information, preferably with a schematic or photo depicting the flow cell setup. The electrolyte, flow rate, gas flow conditions, reference electrode, catalyst loading on the electrode and preparation information etc. should be included.

Reviewer #2 (Remarks to the Author):

The study focuses on the ultra-low coordination number copper catalysts for electrochemical CO₂ reduction, which has good methane selectivity and stability. However, the key argument of this paper, XAS analysis of ultra-low coordinated copper catalysts, looks invalid and incomplete, which needs more explanation and transparency. The ultra-low coordination number obtained from this paper actually indicate the Cu-metal with nano-size, which can be quantitatively estimated with high quality data fitting (A. I. Frenkel, et al. *Annu. Rev. Anal. Chem.* 4, 23–39, 2011; Z. Weng, et al., *Nature Communications*, 9, 415, 2018). Previous report has shown the effective of nano-cluster Cu metal converted from CuPc as the active site for CO₂ reduction (Z. Weng, et al., *Nature Communications*, 9, 415, 2018). I don't see any significant new things that warrant the publishing in Nat Comm.

In addition, I have the following comments for authors to address for the improvement and possible submission to other journals.

1. The author argued that the mixture of CuPc with CNP can help form the metallic Cu with nanosize to catalyze the reaction. It is possible the pure CuPc catalysts authors prepared have low surface-to-bulk ratio. The surface CuPc convert to Cu metals due to the surface reactions. XAS is a bulk

technique that cannot detect the surface changes, thus no formation of Cu is observed. CuPc to CNP with 1:7 ratio was used for electrochemical performance but unfortunately not used for XAS analysis. This part is critical to create the turning point if authors conclusion can be held. I will suggest to add another data point, say CuPc to CNP with 1:10 ratio for both electrochemical and XAS analysis, in addition to 1:7 ratio one for XAS analysis. This is to guarantee the effective of the volcano shape for both electrochemical performance and XAS analysis.

2. Only XAS measurements with controlled current are performed. It will be interesting and more important to perform XAS under voltage-controlled mode, namely at different applied voltage to see the reversible or irreversible changes of CuPc to Cu, similar to what report in previous study (Z. Weng, et al., Nature Communications, 9, 415, 2018). This is to confirm the metallic Cu as the true catalytic site for this reaction.

3. It seems the Method section has been written in a rush, at least it does not allow experimentally reproducing the results presented in this study. Please provide a detailed Method section for XAS experiment and data analysis, such as in-situ cell design and XAS analysis package.

4. Please provide the detailed method to determine all sample compositions by XAS (using XANES or EXAFS? Linear combination fit or linear function analysis?) and cited related reference.

5. The author argued that 4:1 ratio sample stabilized after 50 minutes; however, the XANES and first derivative of XANES do not show any difference between 25 mins and 50 mins. The author should provide the enlarged view of XANES, full XAS spectra, and EXAFS k-space spectra to prove the stable after 50 minutes.

6. The author should provide the method of how to get the Cu-Cu coordination number such as 4.2. The percent of composition cannot determine the Cu-Cu coordination number. EXAFS fitting needs to be done since the author did not mention any analysis method of EXAFS. Suppose the author did run the EXAFS fitting, the fitting table with all necessary fitting parameters (such as amplitude reduction factor, scattering path length, K-range, R-factor, Debye-Waller factor, and the corresponding error bars). The fitting of EXAFS k-space and comparison of EXAFS k-space also need to be provided to see the actual differences.

REVIEWER COMMENTS

Reviewer #1 (Remarks to the Author):

In this article, the authors investigate electrocatalysts for the conversion of CO₂ to CH₄. Specifically, the authors find via computational studies that Cu in a low coordination environment favors the conversion of CO₂ to CH₄ in alkaline media as opposed to multi-C products. The authors then develop a strategy to use carbon nanoparticles to confine Cu to an ultra-low coordination environment in their prepared MEAs. By increasing the amount of C nanoparticles, the authors show that the Cu coordination number can be reduced to 4.2, which hinders the formation of C-C products and promotes CH₄ formation. Such a study of the restructuring of Cu-based site isolated catalysts and its tuning to modulate product distribution is of prime importance for the field, and fully justifies its publication in this journal.

However, this article would require some corrections and additions before the work could be published, mainly regarding the processing and analysis of XAS data. A list of comments and questions can be found below:

Response from authors:

We appreciate the reviewer's suggestions, and we have revised the manuscript and SI based on the specific comments below.

1. Page 6, the authors state "The ratio of 7:1 is the exception to this trend, as all CO₂RR product FEs, including CH₄ and C₂H₄, decreased while the hydrogen evolution increased (Supplementary Fig. 2b). We attribute this selectivity shift to a shortage of Cu reaction sites, suggesting a trade-off between methanation and reduced CO₂RR activity as the ratio of CNP to CuPc is increased" This formulation is not very clear. Are the authors claiming a shift in selectivity originating from a mass-transport limitation of CO₂ to the active sites? This statement may need to be reformulated.

Response from authors:

Increasing the CNP-to-CuPc ratio decreases the Cu active sites and increases the C active sites. The C sites produce predominantly H₂ (Supplementary Fig. 1), whereas the Cu sites are the only ones capable of reducing CO₂ to methane. As the Cu sites become less

prevalent, we expect more H₂ to be generated. We have modified the text to better reflect this:

“As the CNP active sites increased relative to that of the Cu, more H₂ production is expected because CNP active sites cannot perform CO₂RR and instead produce H₂ (Supplementary Fig. 1). A lower density of Cu sites thus lowers CO₂RR activity. This suggested a trade-off between methanation and reduced CO₂RR activity as the ratio of CNP to CuPc was increased.”

2. The XANES and EXAFS for 4:1 CNP:Cu do not seem to agree so well. The XANES indicate that all of the Cu(II) is reduced to Cu(0) within the first 25 minutes and that edge position remains stable afterwards. This is further highlighted by the first derivative (Figure 3c). The EXAFS, on the other hand, show that the Cu(II) is gradually reduced and a significant amount remains after 25 minutes and there is even still some present after 50 minutes (Supp. Figure 3c). In comparison, the XANES for 1:1 CNP:CuPc also seem to be in disagreement. The XANES (Figure 3b) seem to show a gradual shift from Cu(II) to Cu(0) with time while the EXAFS show that most of the Cu(II) has been converted to Cu(0) within the first 20 min and changes only slightly afterwards (SI Figure 3b).

Could the authors explain these discrepancies? Is it possible that the spectra in Figures 3b and 3c were accidentally switched?

Response from authors:

The EXAFS spectra of the 4:1 sample were incorrectly labelled and had since been corrected with the proper data (Fig. 3 and Supplementary Fig. 6); we have also repeated the XANES measurements on the 1:1 sample (Fig. 3 and Supplementary Fig. 4). These updated XANES and EXAFS measurements are now in agreement. These in-situ XAS measurements now show that the CuPc is rapidly transformed to Cu(0) by the time of the first CO₂RR measurement and then stabilised (since the profiles are similar at all sampling times during CO₂RR):

“

Supplementary Figure 4. In-situ sample characterisation under electrocatalytic reaction conditions. Cu K-edge XANES spectra and first-order derivatives of the XANES spectra (collected at 200 mA cm^{-2} under CO₂RR conditions) for the sample containing (a) only CuPc (b) a 1:1 ratio of CNP to CuPc (c) a 4:1 ratio of CNP to CuPc.

Supplementary Figure 6. In-situ sample characterisation under electrocatalytic reaction conditions. Fourier-transformed Cu EXAFS spectra (collected at 200 mA cm^{-2} under CO_2RR conditions) for a sample containing (a) metallic Cu foil, (b) only CuPc, (c) a 1:1 ratio of CNP to CuPc (d) a 4:1 ratio of CNP to CuPc.”

Since the XANES spectra collected at different times during CO₂RR now have similar profiles, we simplified the characterisation figure in the manuscript (Fig. 3) and moved the detailed figure (Supplementary Fig. 4) to Supplementary Information.

“

“Fig. 3 In-situ sample characterisation under electrocatalytic reaction conditions. Cu K-edge XANES spectra and first-order derivatives of the XANES spectra (collected at 200 mA cm⁻² under CO₂RR conditions) for the sample containing (a) only CuPc (b) a 1:1 ratio of CNP to CuPc (c) a 4:1 ratio of CNP to CuPc.”

”

- Figure 3d shows the fitting after CO₂RR (no potential applied)? Or under which applied potential? This should be clarified in the Figure 3 caption.

Response from authors:

The EXAFS measurements were performed at 200 mA/cm². We have now clarified this in the **Figure 3** caption:

“Fig. 3 In-situ sample characterisation under electrocatalytic reaction conditions. Cu K-edge XANES spectra and first-order derivatives of the XANES spectra (collected at 200 mA cm⁻² under CO₂RR conditions) for the sample containing (a) only CuPc (b) a 1:1 ratio of CNP to CuPc (c) a 4:1 ratio of CNP to CuPc. (d) Fourier-transformed Cu K-edge EXAFS spectra (collected at 200 mA cm⁻² under CO₂RR conditions) and fitting lines for samples containing different ratios of CNP to CuPc. (e) Comparison of the metallic Cu-Cu coordination number determined from EXAFS analysis and methanation selectivity.

4. Page 8, lines 1-2, the authors state “The fitted metallic Cu-Cu bond scattering 1 path spectra in Fig. 3d were plotted based on fitting parameters shown in Supplementary Table 1.”

However, this table does not really provide a summary of the fit. What do the CuPC and Cu ratios refer to? This is not explained or described in the text at all. A table of the fitting parameters including the CN, bond length, debye waller factor, e0, r-factor, etc. used or calculated for each scattering path that was included in the fit should be provided for each sample at each potential.

In addition, it appears from that table that the EXAFS fits were carried out assuming only Cu-Cu coordination. Wouldn't the author expect some Cu-C bonds once CNPs are added? If very small Cu particles are considered, then some evidence of Cu-C coordination from the CNPs would be likely, at least from where they are anchored to the CNP surface; could the author comment on that point?

Response from authors:

Supplementary Table 1 contains the XANES-derived compositions. We have added the requested detail regarding the fit into the caption of Supplementary Table 1:

“Supplementary Table 1. The catalyst composition fitting from in-situ XANES sample characterisation. Sample compositions were determined by the linear combination fitting

of X-ray absorption near edge structure (XANES) using copper foil and pure CuPc as the standards in the Athena software¹. The fitting range is from -20 to 30 eV.”

We have provided more details about our EXAFS analysis and curve fitting to the SI. For example, the fitting factors are now described in more detail in the **Supplementary Table 2**. We also clarify in the SI that we used multi-path fitting to consider Cu-C bonds (although none were detected):

“Multi-path fitting was considered for Cu-Cu paths, Cu-N paths (within CuPc molecules), and Cu-C paths (interface between the metallic Cu cluster and CNP). The contribution from Cu-C paths was negligible and the fitting using two paths (Cu-Cu and Cu-N) matched well with the experimental spectra (**Supplementary Fig. 5**). This result confirmed that there were no detectable Cu-C paths formed once CNP was added. The fitting results were listed (shown in **Supplementary Table 2**) using only two-path fitting (Cu-Cu and Cu-N) to clarify these shells in the EXAFS spectra. These fitting results were also consistent with the XANES fitting using linear combination results.

5. Similarly, I found the experimental methods section to be significantly lacking detail regarding XAS data. A better description of the XAS experiments is absolutely needed since these are such an integral part of this study. The fitting method for the EXAFS data needs to be described: Which software was used, what fitting parameters were used, which parameters were varied or held constant, etc. All EXAFS fitting results need to be included and presented in a table (coordination number, bond length, amplitude reduction factor, debye waller factor, E0, r-factor, etc.). Over what range (k and R) were the data fit? Which scattering paths were used to fit the data? At the moment there is nothing to really indicate the quality of the EXAFS fitting. Ideally, an example of one of the fits in k-space and r-space (with the real or imaginary component) should be shown.

Response from authors:

We have added these details, and a table, to the SI. Regarding our XAS experimental measurements and the fitting parameters:

“In-situ XAS was carried out under the same conditions as electrochemical testing using a modified flow cell with an opening in the gas chamber sealed by Kapton tape. The XAS signal was collected by the Vortex detector at the 9BM beamline of Advanced Photon Source (APS) in Argonne National Laboratory and the silicon drift detector at the 44A beamline of National Synchrotron Radiation Research Center (NSRRC). The scan range was kept in an energy range of 8800-9600 eV for the Cu K-edge measurements. The spectra were obtained by subtracting the baseline of the pre-edge and normalising that of the post-edge. EXAFS analysis was conducted using Fourier transform on k^2 -weighted EXAFS oscillations to evaluate the contribution of each bond pair to the Fourier transform peak.

Supplementary Figure 3. Photo of the in-situ XAS characterisation flow-cell electrocatalysis reactor.

REX2000 software using ab initio-calculated phases and amplitudes from the program FEFF 8.2 was used for the EXAFS fitting. The theoretical properties of the Cu-Cu and Cu-N paths, from the crystal information file of metallic Cu and CuPc, were incorporated

to fit the experimental results. The ab initio phases and amplitudes were used in the EXAFS equation:

$$\chi(k) = S_0^2 \sum_j \frac{N_j}{kR_j^2} f_{eff_j}(\pi, k, R_j) e^{-2\sigma_j^2 k^2} e^{-\frac{2R_j}{\lambda_j(k)}} \sin(2kR_j + \phi_{ij}(k))$$

The neighbouring atoms with different distances were divided into j shells. N_j represented the coordination number of shell j at a distance of R_j relative to the central atom. $f_{eff}(\pi, k, R_j)$ was the ab initio amplitude function for shell j , while the Debye–Waller factor $e^{-2\sigma_j^2 k^2}$ accounted for the damping that resulted from static and thermal disorder in absorber–backscatterer distances. The mean free path term $e^{-\frac{2R_j}{\lambda_j(k)}}$ reflected losses due to inelastic scattering, where $\lambda_j(k)$ was the electron mean free path. The sinusoidal term $\sin(2kR_j + \phi_{ij}(k))$, where $\phi_{ij}(k)$ was the ab initio phase function for shell j , reflected the oscillations in the EXAFS. Shake-up/shake-off processes at the central atom(s) affected the amplitude reduction factor, S_0^2 . CN, R , ΔE , and the EXAFS Debye–Waller factor (DW; σ^2) were variable parameters of the EXAFS equation for fitting the experimental result. The R range for fitting ranged from 1.20 to 2.80 and the k fell between 3.70 and 9.55.

Supplementary Table 2. Fitting parameters for the samples using metallic Cu-Cu and Cu-N paths as two-path fitting.

	Paths	N	R (Å)	ΔE (eV)	DW (Å ²)	R-factor
Metallic Cu	Cu-Cu	11.3(2)	2.51(1)	0.8(4)	0.008(2)	0.011
	Cu-N	-	-	-	-	
CuPC	Cu-Cu	9.2(4)	2.54(3)	0.5(4)	0.009(2)	0.014
	Cu-N	0.1(0)	1.83(4)	-1.4(3)	0.007(1)	
C:Cu=1:1	Cu-Cu	6.9(2)	2.54(8)	-1.7(3)	0.009(0)	0.007
	Cu-N	0.2(1)	1.81(3)	3.8(3)	0.005(1)	
C:Cu=4:1	Cu-Cu	4.9(1)	2.53(6)	2.6(6)	0.008(1)	0.006
	Cu-N	1.2(2)	1.84(7)	-6.4(5)	0.006(4)	

Supplementary Figure 5. EXAFS spectra fitting results in R and k space. Cu EXAFS spectra R and k space fitting for the sample containing (a) a metallic Cu foil (b) only CuPc (c) a 1:1 ratio of CNP to CuPc (d) a 4:1 ratio of CNP to CuPc.

- The second main point that currently appears unclear from the article is the timeframe of the transformation of the CuPC to Cu nanoparticles. Could the author develop on that point? In particular, how long do CNP:CuPC samples need to be reduced before they reach optimal selectivity for CH₄? Is there a break in period or some preconditioning

applied where the CO₂RR is increasing as the catalyst is formed? Or are the data in Figure 2c/d collected immediately after assembling the cell?

Response from authors:

Our approach was to compare the performance of the different CNP:CuPc samples at steady state conditions. We waited 2 hours prior to taking gas product samples to accommodate any restructuring. Our XANES data (Fig. 3a-c) suggest relatively rapid transformations, with minimal change in structure beyond the first 80 minutes of the reaction; sampling at 2 hours well-approximated steady-state conditions. We have added a line of clarification in the manuscript:

“For the screening of samples with different CuPc:CNP ratios, gas and liquid samples were taken after two hours of CO₂RR to ensure that the system was at steady state.”

7. In the same line, is there a reason that the XAS results are shown for OCC (i.e. 0 min), 3, 6, and 10 min for the in-situ characterisation of CuPC while for the preparation of 4:1 CNP:CuPC the first data point is after 25 minutes? Is there anything interesting that happens within the first few minutes that shows the formation of the catalyst? From the XANES it seems that all the Cu(II) is reduced on a shorter time scale for the 4:1 sample.

Response from authors:

We repeated the XANES measurements of the 1:1 sample (see updated **Fig. 3** and **Supplemental Fig. 4**) and the new results are more consistent with that of the 4:1 sample (which was previously mislabelled as explained in our response to concern #2): specifically, they both exhibit significant change within the first 20 minutes and minimal change after. It was very challenging to make the in-situ measurements for 1:1 and 4:1 samples because the low copper concentrations gave weak fly scan results. To compensate for the faint signals in these scans, we used longer scan times at the expense of some temporal resolution. Moreover, we were not concerned to analyse the 1:1 and 4:1 at the shorter time scales because we knew, from our prior experiments, that they were relatively stable over the span of hours and we wanted to showcase the coordination stability at longer time scales. We have added some clarification to the text:

“To investigate the structural stability of the lower coordination Cu states, we ran the 1:1 and 4:1 samples for longer times.”

8. Page 8, line 10-12 the authors state “The XAS characterisation employed here yields an average coordination number within the first ~100 nm of the bulk, and thus provides a conservative (e.g. high) estimate of the coordination number at the surface.

I am not sure what is meant by this statement. How big do the authors expect the Cu particles to be? From the post-electrolysis TEM image (supp. Figure 9) it appears they should be around 2-5 nm, so XAS should be very representative of what is happening at the surface. Anything much larger than 20 nm in diameter would only yield information related to the bulk.

Response from authors:

We agree with the reviewer that our XAS measurements should be representative of the surface since our cluster size is only 2-5 nm (Supplementary Fig. 15 & Fig. 16). We have deleted this discussion from the text and mentioned the size discussion in MS and SI:

“STEM/EDX and TEM images taken pre- and post-electrolysis also support the claim that Cu nanoclusters (~2-5 nm) were formed during the reaction (Supplementary Fig. 15 & Fig. 16).

”

“

Supplementary Figure 15. Scanning transmission electron microscopy coupled with energy dispersive X-ray (STEM/EDX) spectroscopy images. All imaging is performed on a 4:1 ratio of CNP to CuPc post-electrolysis sample. (a)STEM image (b) Carbon elemental mapping. (c) Cu elemental mapping.

a

b

Supplementary Figure 16. Transmission electron microscopy (TEM) images. All imaging is performed on 4:1 ratio of CNP to CuPc samples (a) Pre-electrolysis TEM image. (b) Post-electrolysis TEM image.”

9. The reported electrolyte concentration seems rather low (0.05 M KHCO₃). Could the authors comment on this choice?

Response from authors:

Higher concentrations of electrolyte will encourage cation diffusion to the cathode through the anion exchange membrane, which does not have 100% permselectivity to anions. These cations can react with carbonates from dissolved CO₂ to form salt precipitates at the cathode. We have added a line in the text to clarify:

“The low concentration of electrolyte was chosen to minimise potassium cation crossover and subsequent salt formation^{42,43}.”

10. The XPS figures in the SI (Supp. Figure 5) are very difficult to read because of the color choice. The light grey color used to plot the measured signal is nearly impossible to read against the white background. This needs to be updated to a darker color that is easily visible. The total fit envelope of the data also needs to be shown and compared to the measured signal, not just the fits of the deconvoluted peak components.

Response from authors:

We have replotted the figures accordingly: “

Supplementary Figure 9. XPS characterisation of pre- and post-electrolysis samples for deconvolved N 1s peaks. Pre-electrolysis and post-electrolysis XPS spectra for a sample containing a (a) 1:1 ratio of CNP to CuPc (b) 2:1 ratio of CNP to CuPc (c) 4:1 ratio of CNP to CuPc.

Supplementary Figure 10. XPS characterisation of pre- and post-electrolysis samples for deconvoluted Cu 2p peaks. Pre-electrolysis and post-electrolysis XPS spectra for a sample containing a (a) 1:1 ratio of CNP to CuPc (b) 2:1 ratio of CNP to CuPc (c) 4:1 ratio of CNP to CuPc.

»

11. From these XPS data, I would argue that a significant amount of CuPC remains even after electrolysis, considering that the CuPC peaks in the Cu 2p and N-Cu peaks in the N 1s are clearly visible in all samples after electrolysis, and also in the XRD. A quantitative analysis should be applied to the XPS data if possible in order to identify how much CuPC remains after electrolysis. The incomplete conversion of CuPC to CNP:CuPC also seems to be visible in the EXAFS shown in Supp. Figure 3b and 3c (i.e. the peak remaining around 1.5 Å), which may also contribute to the lower overall CNs that are derived. Do the authors have a hypothesis on this point?

Response from authors:

We agree that there is some CuPc left after reaction as supported by our in-situ XANES and ex-situ XPS measurements. We have attempted to quantify the amount of CuPc remaining by normalising the N-Cu peak by the N=C peak, as we now describe in the SI:

Supplementary Table 3 XPS integrated areas for N 1s peaks. The N-Cu peak area normalised by the inert N=C peak area (the latter of which is assumed to not change during the reaction) was used to estimate the amount of CuPc remaining after the reaction.

XPS Sample	N=C Bond Absolute Peak Area	N-Cu Bond Absolute Peak Area	N-Cu Bond Normalised Peak Area	Percentage of N-Cu Bond Remaining
CNP:CuPc = 1:1 Before	5169.953	37777.295	7.307	36.3%
CNP:CuPc = 1:1 After	3308.329	8785.107	2.655	
CNP:CuPc = 2:1 Before	4842.720	18107.610	3.739	27.8%
CNP:CuPc = 2:1 After	4408.851	4583.436	1.039	
CNP:CuPc = 4:1 Before	6024.507	17582.035	2.918	21.7%
CNP:CuPc = 4:1 After	5472.815	3461.007	0.632	

We have changed the manuscript wording to better reflect the remaining CuPc and direct the reader to this analysis:

“This shift confirmed that most of the Cu within the CuPc catalyst was indeed reduced to metallic Cu(0) during CO₂RR, for all sample compositions (**Supplementary Table 1**).”

“The deconvolved N 1s peaks demonstrate the N-Cu bond in the CuPc molecular structure was decomposed irrevocably⁴⁰ (**Fig. 4a**, **Supplementary Fig. 10**, **Supplementary Table 3**).”

We hypothesise that as the CNP ratio is increased, the catalyst ink becomes more conductive and more CuPc sites are believed to be electronically connected, thereby facilitating their conversion during CO₂RR conditions. The EXAFS peak remaining at 1.4 Å is found in metallic copper as well as CuPc and cannot be differentiated easily. We have added an EXAFS spectrum of metallic Cu to the **Supplementary Fig. 6**. to clarify:

“

Supplementary Figure 6. In-situ sample characterisation under electrocatalytic reaction conditions. Fourier-transformed Cu EXAFS spectra (collected at 200 mA cm^{-2} under CO_2RR conditions) for a sample containing (a) metallic Cu foil, (b) only CuPc, (c) a 1:1 ratio of CNP to CuPc (d) a 4:1 ratio of CNP to CuPc.”

12. The authors state that “the higher magnification TEM images taken pre- and post-electrolysis also support the claim that no sizable metallic Cu nanoparticles were formed during the reaction while also suggesting that the metallic Cu was in an amorphous state during electrolysis.”

What do the author consider sizeable metallic Cu nanoparticles? Based on the comparison of the TEM images, it seems that some particles in the range of 2-5 nm could be suspected. I recommend adding STEM and EDX mapping to the paper to help with identifying the presence or absence of small Cu nanoparticles. Being able to map the distribution of Cu before and after electrolysis would be highly desirable.

Response from authors:

We have now performed both SEM with EDX mapping, and STEM with EDX mapping on a sample after electrolysis:

“

Supplementary Figure 13. Scanning electron microscopy coupled with energy dispersive X-ray (SEM/EDX) spectroscopy images. All imaging is performed on a 4:1 ratio of CNP to CuPc post-electrolysis sample. (a) Carbon elemental mapping. (b) Cu elemental mapping.

Supplementary Figure 14. SEM/EDX spectra results performed on a 4:1 ratio of CNP to CuPc post-electrolysis sample.

Supplementary Table 4. SEM/EDX spectroscopy element analysis on a 4:1 ratio of CNP to CuPc post-electrolysis sample.

Element spectrum	Spectrum
C	96.73%
F	2.52%
K	0.63%
Cu	0.12%
Total	100.00%

Supplementary Figure 15. Scanning transmission electron microscopy coupled with energy dispersive X-ray (STEM/EDX) spectroscopy images. All imaging is performed on a 4:1 ratio of CNP to CuPc post-electrolysis sample. (a)STEM image. (b) Carbon elemental mapping. (c) Cu elemental mapping.

We mention these results in the manuscript:

“Energy dispersive X-ray (EDX) mapping and spectroscopy results proved the Cu element is evenly distributed on the GDE (Supplementary Fig. 13 & Fig. 15 & Supplementary Table 4). STEM/EDX and TEM images taken pre- and post-electrolysis also support the claim that Cu nanoclusters (~2-5 nm) were formed during the reaction (Supplementary Fig. 15 & Fig. 16).”

Some additional minor issues:

13. In the SI Figure 2 caption, “(b) The Faradaic Efficiency (FE) toward hydrogen with different CNP to CuPc ratios.” is written twice.

Response from authors:

Thank you for pointing this out, we have now removed the duplicate reference:

“

(b) The Faradaic Efficiency (FE) toward hydrogen with different CNP to CuPc ratios. (c) The FE toward methane (CH₄) with different CNP to CuPc ratios. (d) The FE toward ethylene with different CNP to CuPc ratios. (e) The FE toward carbon monoxide with different CNP to CuPc ratios. (f) The FE of total CO₂RR products with different CNP to CuPc ratios.

”

14. Supp. Figure 11 is not referenced in the article despite its interesting content and should be mentioned in the main text.

Response from authors:

We now call out this figure in the main text:

“The highest FE toward CH₄, 62%, was exhibited by the 4:1 sample at -4.00 V and 220 mA cm⁻² (liquid product analysis shown in Supplementary Fig. 2).”

15. Are the XANES and EXAFS shown in Figure 3 performed in a saturated electrolyte (i.e. under constant CO₂ bubbling)? This should be indicated in the figure caption.

Response from authors:

As noted in response #5, we provide full details on the XANES and EXAFS measurements, and describe in the SI that we performed these measurements in a flow cell with a gas diffusion electrode, and as such CO₂ bubbling into electrolyte was not required.

16. A better description of the flow cell setup should be added in the experimental part or supplementary information, preferably with a schematic or photo depicting the flow cell setup. The electrolyte, flow rate, gas flow conditions, reference electrode, catalyst loading on the electrode and preparation information etc. should be included.

Response from authors:

We have added these details to the experimental section and directed the reader to this section in the main text:

“

Prepared GDEs were coupled with an anion exchange membrane and iridium oxide-based anode for oxygen evolution in the MEA (**Methods, Electrode Preparation & Electrochemical reduction of CO₂**).

”

“

All CO₂RR experiments were performed using a MEA electrolyser **with an active area of 5 cm² (Supplementary Fig. 18)**. During a CO₂RR experiment, the aqueous 0.05 M KHCO₃ **anolyte** was circulated through the anode flow channel **at a flow rate of 25 mL/min using a peristaltic pump**. An anion exchange membrane (Sustainion X37-50, **Dioxide Materials**) was used as the solid cathode electrolyte. The CO₂ gas flow rate, **supplied at a rate of 80 standard cubic centimetres per minute (sccm), was bubbled through water for humidification prior to entering the electrolyser. All voltages reported are full cell voltages without *i*R compensation.**

”

“

a

b

Supplementary Figure 17. Membrane electrode assembly (MEA) CO₂ electrolyzer. (a) Schematic of the MEA cell. (b) Photo of the MEA cell and associated system equipment.”

Reviewer #2 (Remarks to the Author):

The study focuses on the ultra-low coordination number copper catalysts for electrochemical CO₂ reduction, which has good methane selectivity and stability. However, the key argument of this paper, XAS analysis of ultra-low coordinated copper catalysts, looks invalid and incomplete, which needs more explanation and transparency. The ultra-low coordination number obtained from this paper actually indicate the Cu-metal with nano-size, which can be quantitatively estimated with high quality data fitting (A. I. Frenkel, et al. Annu. Rev. Anal. Chem. 4, 23–39, 2011; Z. Weng, et al., Nature Communications, 9, 415, 2018). Previous report has shown the effective of nanocluster Cu metal converted from CuPc as the active site for CO₂ reduction (Z. Weng, et al., Nature Communications, 9, 415, 2018). I don't see any significant new things that warrant the publishing in Nat Comm.

In addition, I have the following comments for authors to address for the improvement and possible submission to other journals.

Response from authors:

We appreciate the reviewer's perspective and follow-up on the characterisation concern and the novelty point, with new measurements and clarifications in the text as described in-line below.

1. The author argued that the mixture of CuPc with CNP can help form the metallic Cu with nanosize to catalyze the reaction. It is possible the pure CuPc catalysts authors prepared have low surface-to-bulk ratio. The surface CuPc convert to Cu metals due to the surface reactions. XAS is a bulk technique that cannot detect the surface changes, thus no formation of Cu is observed. CuPc to CNP with 1:7 ratio was used for electrochemical performance but unfortunately not used for XAS analysis. This part is critical to create the turning point if authors conclusion can be held. I will suggest to add another data point, say CuPc to CNP with 1:10 ratio for both electrochemical and XAS analysis, in addition to 1:7 ratio one for XAS analysis. This is to guarantee the effective of the volcano shape for both electrochemical performance and XAS analysis.

Response from authors:

As the reviewer requested, we attempted to characterise the 7:1 sample but the very low copper concentration in this sample made measurements incredibly challenging. We used

the silicon drift detector, the most sensitive detector available to us, but we could not get reliable signals. The 10:1 sample would be even more difficult to characterise as the copper concentrations would be even lower.

2. Only XAS measurements with controlled current are performed. It will be interesting and more important to perform XAS under voltage-controlled mode, namely at different applied voltage to see the reversible or irreversible changes of CuPc to Cu, similar to what report in previous study (Z. Weng, et al., Nature Communications, 9, 415, 2018). This is to confirm the metallic Cu as the true catalytic site for this reaction.

Response from authors:

We thank the reviewer for highlighting this prior work which studied, but did not try to control, Cu agglomeration from CuPc. This paper does not present any experiments or characterisations taken at different times, so the stability of their synthesis approach is not known. Our work builds on this by developing a means to stabilise these low coordination numbers for methane production.

As the reviewer requested, we have now performed a set of XANES and EXAFS measurements with the 4:1 sample at different constant voltages as requested (**Supplementary Fig. 8**). Unlike the referenced paper, we observe only small changes in CuPc transformation for the more negative applied voltages, at all voltages most of the CuPc is transformed into Cu, emphasising that the CNP moderator strategy can effectively limit Cu agglomeration. We now mention these results in the manuscript:

“Potentiostatic XAS measurements suggested that CuPc agglomeration was not influenced significantly by the applied potential when the CNP moderator strategy was employed (**Supplementary Fig. 8**).”

Supplementary Figure 8. In-situ XAS characterisation under electrocatalytic reaction conditions, with voltage range from -1.0 V to -2.0 V vs. RHE (non-*i*R corrected), for a 4:1 ratio CNP to CuPc sample. (a) Cu XANES spectra. (b) Fourier-transformed Cu EXAFS spectra.

- It seems the Method section has been written in a rush, at least it does not allow experimentally reproducing the results presented in this study. Please provide a detailed Method section for XAS experiment and data analysis, such as in-situ cell design and XAS analysis package.

Response from authors:

We now provide a comprehensive XAS analysis method section in SI, including the cell design and the XAS analysis approach.

In-situ X-ray absorption spectroscopy (XAS) Characterization

In-situ XAS was carried out under the same conditions as electrochemical testing using a modified flow cell with an opening in the gas chamber sealed by Kapton tape. The XAS signal was collected by the Vortex detector at the 9BM beamline of Advanced Photon Source (APS) in Argonne National Laboratory and the silicon drift detector at the 44A beamline of National Synchrotron Radiation Research Center (NSRRC). The scan range was kept in an energy range of 8800-9600 eV for the Cu K-edge measurements. The spectra were obtained by subtracting the baseline of the pre-edge and normalising that of the post-edge. K-edge extended X-ray absorption fine structure (EXAFS) analysis was conducted using Fourier transform on k^2 -weighted EXAFS oscillations to evaluate the contribution of each bond pair to the Fourier transform peak.

Supplementary Figure 3. Photo of the in-situ XAS characterisation flow-cell electrocatalysis reactor.

REX2000 software using ab initio-calculated phases and amplitudes from the program FEFF 8.2 was used for the EXAFS fitting. The theoretical properties of the Cu-Cu and Cu-N paths, from the crystal information file of metallic Cu and CuPc, were incorporated to fit the experimental results. The ab initio phases and amplitudes were used in the EXAFS equation:

$$\chi(k) = S_0^2 \sum_j \frac{N_j}{kR_j^2} f_{effj}(\pi, k, R_j) e^{-2\sigma_j^2 k^2} e^{-\frac{2R_j}{\lambda_j(k)}} \sin(2kR_j + \phi_{ij}(k))$$

The neighbouring atoms with different distances were divided into j shells. N_j represented the coordination number of shell j at a distance of R_j relative to the central atom. $f_{eff}(\pi, k, R_j)$ was the ab initio amplitude function for shell j , while the Debye–Waller factor $e^{-2\sigma_j^2 k^2}$ accounted for the damping that resulted from static and thermal disorder in absorber–backscatterer distances. The mean free path term $e^{-\frac{2R_j}{\lambda_j(k)}}$ reflected losses due to inelastic scattering, where $\lambda_j(k)$ was the electron mean free path. The sinusoidal term $\sin(2kR_j + \phi_{ij}(k))$, where $\phi_{ij}(k)$ was the ab initio phase function for shell j , reflected the oscillations in the EXAFS. Shake-up/shake-off processes at the central atom(s) affected the amplitude reduction factor, S_0^2 . CN, R , ΔE , and the EXAFS Debye–Waller factor (DW; σ^2) were variable parameters of the EXAFS equation for fitting the experimental result. The R range for fitting ranged from 1.20 to 2.80 and the k fell between 3.70 and 9.55.

Supplementary Table 2. Fitting parameters for the samples using metallic Cu-Cu and Cu-N paths as two-path fitting.

	Paths	N	R (\square)	ΔE (eV)	DW (\square^2)	R-factor
Metallic Cu	Cu-Cu	11.3(2)	2.51(1)	0.8(4)	0.008(2)	0.011
	Cu-N	-	-	-	-	
CuPC	Cu-Cu	9.2(4)	2.54(3)	0.5(4)	0.009(2)	0.014
	Cu-N	0.1(0)	1.83(4)	-1.4(3)	0.007(1)	
C:Cu=1:1	Cu-Cu	6.9(2)	2.54(8)	-1.7(3)	0.009(0)	0.007
	Cu-N	0.2(1)	1.81(3)	3.8(3)	0.005(1)	
C:Cu=4:1	Cu-Cu	4.9(1)	2.53(6)	2.6(6)	0.008(1)	0.006
	Cu-N	1.2(2)	1.84(7)	-6.4(5)	0.006(4)	

Multi-path fitting was considered for Cu-Cu paths, Cu-N paths (within CuPc molecules), and Cu-C paths (interface between the metallic Cu cluster and CNP). The contribution from Cu-C paths was negligible and the fitting using two paths (Cu-Cu and Cu-N) matched well with the experimental spectra (**Supplementary Fig. 5**). This result confirmed that there were no detectable Cu-C paths formed once CNP was added. The fitting results were listed (shown in **Supplementary Table 2**) using only two-path fitting (Cu-Cu and Cu-N) to clarify these shells in the EXAFS spectra. These fitting results were also consistent with the XANES fitting using linear combination results.

Supplementary Figure 5. EXAFS spectra fitting results in R and k space. Cu EXAFS spectra R and k space fitting for the sample containing (a) a metallic Cu foil (b) only CuPc (c) a 1:1 ratio of CNP to CuPc (d) a 4:1 ratio of CNP to CuPc.

»

4. Please provide the detailed method to determine all sample compositions by XAS (using XANES or EXAFS? Linear combination fit or linear function analysis?) and cited related reference.

Response from authors:

We have added this description to the caption of **Supplemental Table 1** regarding our XANES fitting:

“Supplementary Table 1. The catalyst composition fitting from in-situ XANES sample characterisation. Sample compositions were determined by the linear combination fitting of X-ray absorption near edge structure (XANES) using copper foil and pure CuPc as the standards in the Athena software¹. The fitting range is from -20 to 30 eV.”

5. The author argued that 4:1 ratio sample stabilized after 50 minutes; however, the XANES and first derivative of XANES do not show any difference between 25 mins and 50 mins. The author should provide the enlarged view of XANES, full XAS spectra, and EXAFS k-space spectra to prove the stable after 50 minutes.

Response from authors:

The original manuscript was not sufficiently clear here. We have now clarified that the sample was stable *by* 50 minutes of operation (and not after):

“The EXAFS spectra of the 4:1 ratio sample stabilised within 50 minutes of operation with little change between the spectra at 50 and 80 minutes, demonstrating that the structure was stable once agglomerated (Supplementary Fig. 3c)”

We have corrected the EXAFS measurements of the 4:1 sample (which were originally mislabelled in error). The XANES and EXAFS measurements of the 4:1 sample now show similar spectra for all times during CO₂RR, suggesting a stable structure.

“

Supplementary Figure 4. In-situ sample characterisation under electrocatalytic reaction conditions. Cu K-edge XANES spectra and first-order derivatives of the XANES spectra (collected at 200 mA cm⁻² under CO₂RR conditions) for the sample containing (a) only CuPc (b) a 1:1 ratio of CNP to CuPc (c) a 4:1 ratio of CNP to CuPc.

Supplementary Figure 6. In-situ sample characterisation under electrocatalytic reaction conditions. Fourier-transformed Cu EXAFS spectra (collected at 200 mA cm^{-2} under CO_2RR conditions) for a sample containing (a) metallic Cu foil, (b) only CuPc, (c) a 1:1 ratio of CNP to CuPc (d) a 4:1 ratio of CNP to CuPc.

»

6. The author should provide the method of how to get the Cu-Cu coordination number such as 4.2. The percent of composition cannot determine the Cu-Cu coordination number. EXAFS fitting needs to be done since the author did not mention any analysis method of EXAFS. Suppose the author did run the EXAFS fitting, the fitting table with all necessary fitting parameters (such as amplitude reduction factor, scattering path length, K-range, R-factor, Debye-Waller factor, and the corresponding error bars). The fitting of EXAFS k-space and comparison of EXAFS k-space also need to be provided to see the actual differences.

Response from authors:

As described above in response #3, we now provide the full details on our XANES and EXAFS measurements and associated fittings in the manuscript and SI.

We thank the reviewers again for their engagement with this work and constructive input.

Reviewer #1 (Remarks to the Author):

The authors have now answered all my initial concerns, and I would suggest to accept the current manuscript for publication in Nature Communication.

Reviewer #4 (Remarks to the Author):

In this manuscript, the authors study the electrochemical conversion of CO₂ to methane. Motivated by their computational study, which shows favorable selectivity for methane on ultra-low coordination Cu, authors perform detailed experiments and characterization. This work is recommended for publication in Nature Communications after the authors address my concerns.

1. Based on the current understandings, authors clearly identify the two selectivity controlling steps for methane formation (electrochemical hydrogenation to *CHO) and for C₂ products (*CO coupling with another *CO to produce *OCCO in a purely chemical manner where there is no electron transfer is involved). However, in the supporting information reaction energies for C-C coupling pathway are calculated by *CO + *CO + H⁺ + e⁻ → *OCCOH. This reaction is clearly not a chemical step as it involves a proton transfer. Please clarify. Instead of calculating reaction energies for *CO + *CO + H⁺ + e⁻ → *OCCOH, the authors should report the values for *CO + *CO + → *OCCO + * in Figure 1d.

2. In the DFT study, the authors have used the standard coordination number of Cu active sites. However, it is well established (Calle-Vallejo et al. Science 2015, 350 (6257), 185–189) that the active sites on a transition metal nanoparticle of different sizes have the same coordination number but correspond to varying adsorption energies. Therefore, the authors should consider generalized coordination number (GCN) which combines the first and second nearest neighbors, and capture the structure sensitivity more accurately. I suggest reporting the results in Figures 1c and 1d as a function of GCN.

3. Hydrogen evolution reaction (HER) is the key side reaction that competes with eCO₂RR. The authors should consider the H* adsorption on these active site moieties and explain how this correlates with the observed favorable selectivity towards methane.

Below are some other remarks, corrections, and suggestions about the manuscript.

1. How did the authors determine the surface area of different configurations in Supplementary Table 5 to calculate the surface energy? Is it a fixed surface area based on the unit cell?

2. I suggest reporting vacancy formation energies as it would be a better descriptor than reporting surface energies.

3. Please indicate the active site (by highlighting the atom/ using a different color) and include side configurations in Supplementary Table 5. In the current state, it is hard to determine what is the active site.

REVIEWER COMMENTS

Reviewer #1 (Remarks to the Author):

The authors have now answered all my initial concerns, and I would suggest to accept the current manuscript for publication in Nature Communication.

We appreciate the reviewer's constructive perspectives.

Reviewer #4 (Remarks to the Author):

In this manuscript, the authors study the electrochemical conversion of CO₂ to methane. Motivated by their computational study, which shows favorable selectivity for methane on ultra-low coordination Cu, authors perform detailed experiments and characterisation. This work is recommended for publication in Nature Communications after the authors address my concerns.

We appreciate the reviewer's suggestions, and we have revised the manuscript and SI based on the specific comments below.

1. Based on the current understandings, authors clearly identify the two selectivity controlling steps for methane formation (electrochemical hydrogenation to *CHO) and for C₂ products (*CO coupling with another *CO to produce *OCCO in a purely chemical manner where there is no electron transfer is involved). However, in the supporting information reaction energies for C-C coupling pathway are calculated by *CO + *CO + H⁺ + e⁻ → *OCCOH. This reaction is clearly not a chemical step as it involves a proton transfer. Please clarify. Instead of calculating reaction energies for *CO + *CO + H⁺ + e⁻ → *OCCOH, the authors should report the values for *CO + *CO + → *OCCO + * in Figure 1d.

We have corrected the reaction as noted, and re-run the simulations. We now simulate the chemical step instead of the electrochemical one. This has been corrected in Figure 1d and in the SI:

MS:

The reaction energies for *CO to form an *OCCO intermediate via C-C coupling did not change significantly with coordination number.

Fig. 1 CO₂RR methanation strategy and DFT calculation. **a** Schematic of electrochemical carbon recycling in a membrane electrode assembly (MEA)-based electrolyser for CO₂-to-CH₄. **b** Schematic of key reaction pathways for CO₂RR: hydrogenation to *CHO for CH₄ production and C-C coupling to *OCCO leading to C₂ generation. **c** Reaction energies for *CO hydrogenation to *CHO on Cu catalysts of various generalised coordination numbers. **d** Reaction energies for *CO coupling to *OCCO on Cu catalysts of various generalised coordination numbers.

SI:

Similarly, the C-C coupling process was simulated by:

2. In the DFT study, the authors have used the standard coordination number of Cu active sites. However, it is well established (Calle-Vallejo et al. Science 2015, 350 (6257), 185–189) that the active sites on a transition metal nanoparticle of different sizes have the same coordination number but correspond to varying adsorption energies. Therefore, the authors should consider generalised coordination number (GCN) which combines the first and second nearest neighbors, and capture the structure sensitivity more accurately. I suggest reporting the results in Figures 1c and 1d as a function of GCN.

We have converted the coordination number to GCN, and updated Figure 1c and 1d, accordingly. The details of GCN are now shown in Supplementary Table 5.

MS:

Fig. 2 CO₂RR methanation strategy and DFT calculation. **a** Schematic of electrochemical carbon recycling in a membrane electrode assembly (MEA)-based electrolyser for CO₂-to-CH₄. **b** Schematic of key reaction pathways for CO₂RR: hydrogenation to *CHO for CH₄ production and C-C coupling to *OCCO leading to C₂ generation. **c** Reaction energies for *CO hydrogenation to *CHO on Cu catalysts of various generalised coordination numbers. **d** Reaction energies for *CO coupling to *OCCO on Cu catalysts of various generalised coordination numbers.

SI:

Supplementary Table 5. The coordination number (CN) and generalised coordination number (GCN) of different configurations

Configurations	$\sum CN$	CN_{max}	GCN
----------------	-----------	------------	-----

	$6*9+3*12$	12	7.5
	$2*8+3*9+11+2*12$	12	6.5
	$2*8+2*9+10+2*12$	12	5.67
	$3*8+3*11$	11	5.18
	$2*8+10+2*11$	11	4.36
	$7+2*10+11$	11	3.45
	$3*10$	10	3

3. Hydrogen evolution reaction (HER) is the key side reaction that competes with eCO₂RR. The authors should consider the H* adsorption on these active site moieties and explain how this correlates with the observed favorable selectivity towards methane.

We have run the *H adsorption simulation and added *H adsorption energy results in Supplementary fig. 1 and highlight the results in the text:

MS:

The reaction energies for *CO to form an *OCCO intermediate via C-C coupling did not change significantly with coordination number. Similarly, the adsorption of H⁺ to *H (on the pathway to hydrogen evolution) is not significantly influenced by the low coordination states when compared to higher coordination states (Supplementary Fig. 1).

SI:

Supplementary Fig. 1 CO₂RR methanation DFT calculation. Reaction energies for *H adsorption on Cu catalysts of various generalised coordination numbers

Below are some other remarks, corrections, and suggestions about the manuscript.

1. How did the authors determine the surface area of different configurations in Supplementary Table 5 to calculate the surface energy? Is it a fixed surface area based on the unit cell?

In the previous manuscript, the surface energy was calculated using the area of the slab model. We have since replaced it with the vacancy formation energy as suggested by reviewer.

SI:

Vacancy formation energy calculation

The vacancy formation energies of various configurations, shown in **Supplementary Table 6**, were defined as,

$$E_v = \frac{1}{n} (E_{defect} - E_{no-defect} + n * E_{Cu})$$

where E_v is the vacancy formation energy of modified Cu configuration, n is the number of Cu vacancies, E_{defect} and $E_{no-defect}$ represent the total energies of the defective and perfect model of the same component, and E_{Cu} is the energy of single Cu atom.

2. I suggest reporting vacancy formation energies as it would be a better descriptor than reporting surface energies.

We have done vacancy formation energy calculations and replaced the surface energy results in Supplementary table 6.

SI:

Supplementary Table 6. Vacancy formation energies of Cu surface models with various coordination numbers

Global Coordination Numbers (GCN)	Configurations	Vacancy formation energy (eV)
7.5		-
6.5		0.34
		0.23
5.67		0.33
		0.26
		0.11
		0.26
		0.28
5.18		0.29
		0.27
		0.26

		0.28
		0.05
		0.24
		0.05
4.36		0.10
		0.15
		0.01
		0.02
3.45		0.13
		0.24
3		0.16

3. Please indicate the active site (by highlighting the atom/ using a different color) and include side configurations in Supplementary Table 5. In the current state, it is hard to determine what is the active site.

We have highlighted the active site in dark red and provided side configurations in Supplementary Table 6.

Supplementary Table 6. Vacancy formation energies of Cu surface models with various coordination numbers

Generalised Coordination Numbers (GCN)	Configurations	Vacancy formation energy (eV)
7.5		-
6.5		0.34
		0.23
5.67		0.33
		0.26
		0.11
		0.26

		0.28
5.18		0.29
		0.27
		0.26
		0.28
		0.05
		0.24
		0.05
4.36		0.10
		0.15
		0.01
		0.02

3.45

0.13

0.24

3

0.16